# Quantitative Evaluation of Color, Firmness, and Soluble Solid Content of Korla Fragrant Pears via IRIV and LS-SVM

**Yuanyuan Liu [1,2,*], Tongzhao Wang [1,2], Rong Su [1,2], Can Hu [1,2], Fei Chen [1,2] and Junhu Cheng [3]**

[1] College of Mechanicaland Electrical Engineering, Tarim University, Alar 843300, China; Wtz080921@163.com (T.W.); 10757202211@stumail.taru.edu.cn (R.S.); 120140004@taru.edu.cn (C.H.); 10757192126@stumail.taru.edu.cn (F.C.)

[2] Agricultural Engineering Key Laboratory at Universities of Education Department of Xinjiang Uygur Autonomous Region, Tarim University, Alar 843300, China

[3] College of Light Industry and Food Sciences, South China University of Technology, Guangzhou 510641, China; fechengjh@scut.edu.cn

* Correspondence: 120080015@taru.edu.cn

**Abstract:** Customers pay significant attention to the organoleptic and physicochemical attributes of their food with the improvement of their living standards. In this work, near infrared hyperspectral technology was used to evaluate the one-color parameter, a*, firmness, and soluble solid content (SSC) of Korla fragrant pears. Moreover, iteratively retaining informative variables (IRIV) and least square support vector machine (LS-SVM) were applied together to construct evaluating models for their quality parameters. A set of 200 samples was chosen and its hyperspectral data were acquired by using a hyperspectral imaging system. Optimal spectral preprocessing methods were selected to obtain out partial least square regression models (PLSRs). The results show that the combination of multiplicative scatter correction (MSC) and Savitsky-Golay (S-G) is the most effective spectral preprocessing method to evaluate the quality parameters of the fruit. Different characteristic wavelengths were selected to evaluate the a* value, the firmness, and the SSC of the Korla fragrant pears, respectively, after the 6 iterations. These values were obtained via IRIV and the reverse elimination method. The correlation coefficients of the validation set of the a* value, the firmness, and the SSC measure 0.927, 0.948, and 0.953, respectively. Furthermore, the values of the regression error weight, $\gamma$, and the kernel function parameter, $\sigma^2$, for the same parameters measure ($8.67 \times 10^4$, $1.21 \times 10^3$), ($1.45 \times 10^4$, $2.93 \times 10^4$), and ($2.37 \times 10^5$, $3.80 \times 10^3$), respectively. This study demonstrates that the combination of LS-SVM and IRIV can be used to evaluate the a* value, the firmness, and the SSC of Korla fragrant pears to define their grade.

**Keywords:** IRIV; LS-SVM; Korla fragrant pear; quality parameter; evaluation

## 1. Introduction

Korla fragrant pears are very popular among customers due to their thin skin, juicy, sweaty taste, and delicate flesh [1,2]. Nowadays, customers pay significant attention to both the organoleptic and physicochemical attributes of fruits with the improvement of their living standards. The organoleptic parameter, color of skin, is related to maturity of Korla fragrant pear. The sunward side of most mature Korla fragrant pears has blush which is also distinctive in all kinds of pears. However, only physicochemical parameters are used as quality evaluation attributes to grade Korla fragrant pears.

Several non-destructive studies have been carried out to evaluate the soluble solid content (SSC) of Korla fragrant pears [3,4]. Zhu et al. [5] used hyperspectral imaging and support vector regression to define this parameter. The correlation coefficient ($R_C$) and the root mean square error ($RMSE_C$) in their calibration set measured 0.986 and 0.186%, respectively. In their validation set the correlation coefficient ($R_V$) and the root mean square error ($RMSE_V$) measured 0.946 and 0.403%. Zhan et al. [6] quantitatively determined the

SSC of Korla fragrant pears via least square support vector machine (LS-SVM) and partial least square regression (PLSR). The $R_V$ and $RMSE_V$ reported in this study measure 0.851 and 0.291%, respectively.

Other researchers investigated the firmness of Korla fragrant pears vie quantitative predictions. For instance, Sheng et al. [7] used near-Infrared (NIR) spectroscopy together with different variable selecting methods to construct a set of partial least square models to describe firmness. Yu et al. [8] predicted both the firmness and the SSC by developing a deep learning method based on Vis/NIR hyperspectral reflectance imaging. Their combination model of a series of stacked auto-encoders and a fully connected neural network achieved a reasonable prediction performance with $R_V$ and $RMSE_V$ values of 0.9434 and 1.81 N, respectively.

However, no investigation reported results on the simultaneous measurement of the organoleptic and physicochemical attributes on Korla fragrant pears. According to the requirements of the latest group standard on Korla fragrant pears [9], organoleptic and physicochemical attributes appear to have the same importance in the grade definition. The skin color of Korla fragrant pears changes from green to red-yellow, as the fruit ripes. The a* value represents the color change from red to green in chromatic aberration data. Therefore, the organoleptic quality of the samples can be defined according to their a* values. The firmness and the SSC are the most significant edible quality parameters in Korla fragrant pears, and they are directly related to consumers' satisfaction [10]. Thus, the three parameters, a*, firmness, and SSC, must be carefully evaluated to determine the influence of the postharvest storage period on the fruit quality control process.

Both quality and safety parameters can be accurately evaluated via hyperspectral imaging [11–13], although the hyperspectral approach requires expensive equipment and complex data analysis Compared with other nondestructive testing methods [14]. However, in order to define simple predicting models and improve their prediction efficiency, a set of wavebands have to be selected. These wavebands can be related with several important chemical bonds, which can be used to discriminate the samples based on their quality and safety parameters. Successive projection algorithms (SPAs) [15–18], competitive adaptive reweighting sampling (CARS) [19,20], and uninformative variable elimination (UVE) [21,22] have been used by to choose such wavebands. Despite these selection methods are quite effective, they do not account for the combination effects among the wavebands. The iteratively retaining informative variables (IRIV) method ensures that each variable has the same probability to take part into the selection process and increases filtering speed by using a set of binary mixing filters [23,24].

To this date, the combination of IRIV and LS-SVM has not been investigated to quantitatively predict the quality parameters of Korla fragrant pears. In this work, IRIV-LS-SVM is used to (1) obtain the a* value, the firmness, and the SSC of Korla fragrant pears, (2) analyze the spectral features of Korla fragrant pears in the 945–1670 nm wavelength range, (3) select the optimal wavebands related to the C-H, N-H, and O-H chemical bonds, and (4) construct a set of predicting models to define the quality parameters for Korla fragrant pears.

## 2. Materials and Methods

### 2.1. Korla Fragrant Pears and Pretreatment

Korla fragrant pears were collected from a plantation located near Tarim University (80°30′–81°58′ E, 40°22′–40°57′ N) from September 11th to September 15th 2019. A set of 200 samples with a uniform shape, a single fruit weight of 120 ± 10 g, and intact epidermis was selected. The side of each Korla fragrant pear, which was exposed to the sunlight, was labeled.

The samples were sprayed with a special fruit cleaning agent (Almawin, Germany), soaked in water for about 30 s, and then rinsed with distilled water twice. The cleaned pears were dried at room temperature (20 °C), and then stored in a preservation box at

4 °C. The samples were placed on the desk at room temperature for 30 min to eliminate the influence of the temperature change before the hyperspectral image data acquisition.

### 2.2. Hyperspectral Imaging System and Diffuse Reflectance Spectrum Data Acquisition

The hyperspectral imaging system used in this study is shown in Figure 1. It consists of a push-broom scanning system composed of a spectrograph (N17E, Spectral Imaging Ltd., Oulu, Finland), an enhanced near-infrared hyperspectral camera (Xeva-1.7-320, Xenics Infrared Solutions, Leuven, Belgium), four halogen light sources with a maximum power of 150 W each, a stepper-motor-driving stage, a dark box, and a computer.

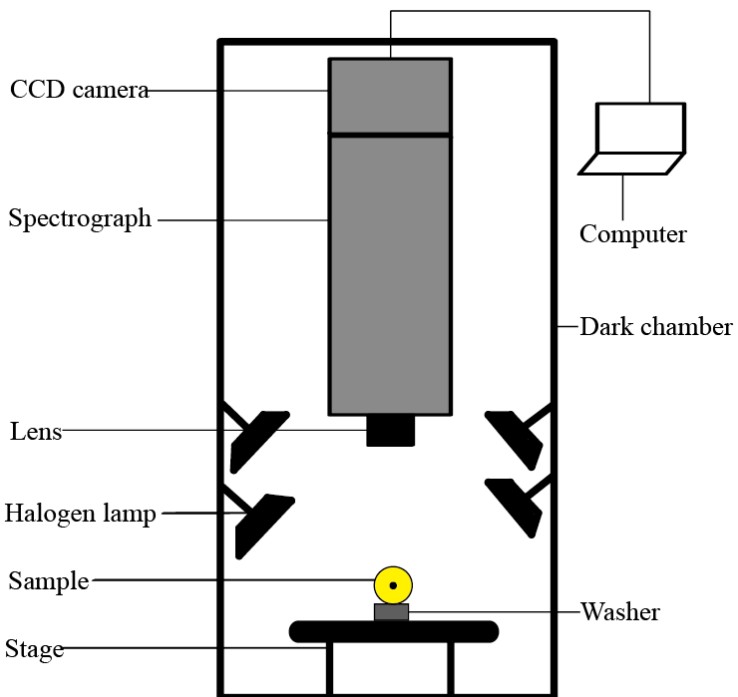

**Figure 1.** Schematic diagram of the hyperspectral imaging system.

Before data acquisition, the system was preheated for about 30 min to ensure its temperature stability. A sample with a uniform shape and a moderate weight was placed on the stage to adjust the calibration parameters of the instrument. In order to ensure the hyperspectral image integrity of the samples, the distance between the bottom of the spectrograph and the stage was set to 310 mm and maintained fixed. Moreover, the focal length was adjusted by rotating the aperture until sharp reflection peaks appeared. The moving speed of the platform and the exposure time of the camera were set to 18 mm/s and 20 ms, respectively.

A strip-shaped standard reflecting whiteboard was placed under the spectrograph to obtain the white and the black references separately by opening and closing the lens cover. The long axis of a Korla fragrant pear sample was positioned along the moving direction of the stage to ensure a uniform irradiation. The sunward side of each Korla fragrant pear was placed upside in order to reduce the influences on spectral data and measured quality parameter values of the sunward side and nightside. The sample hyperspectral image data were corrected by taking into account the black and white references to eliminate the influence of the light source intensity differences and the camera dark current noise, as described in Formula (1).

$$I = (I_o - I_b)/(I_w - I_b) \tag{1}$$

Here, $I_o$ corresponds to the original hyperspectral image data, $I_b$ to the black reference data collected when the lens cover is closed, $I_w$ refers to the white reference image data of the strip-shaped standard reflecting whiteboard when the lens cover is open.

The spectral data of the region of interest (ROI) were extracted by using the ENVI 5.1 software (Exelis Visual Information Solutions, Boulder, Colorado, USA). The shape of the ROI was rectangular, and its center was located near the intersection between the long axis and the equator of the pear. The corresponding pixel numbers of each ROI were 90 along the long axis and 70 along the equator.

### 2.3. Measurement of the Sensory and Physicochemical Parameters

The sensory and physicochemical parameters of the Korla pears were measured after the hyperspectral image data acquisition. The sensory parameter a* was obtained by employing a precision chromatic aberration meter (HP-C220, Shenzhen HanPu Testing Instrument Co., Ltd., Shenzhen, China). Each measurement consisted of an average of five points randomly selected on the ROI surface.

The firmness was obtained by averaging the values collected at five different locations of the pears. They were set at 12 mm center distance between two adjacent ROIs and were measured by a firmness tester (GY-4 with a probe diameter of 7.9 mm, Top instrument). The SSC was measured by using a digital refractometer (PAL-1, ATAGO, Tokyo, Japan). Before the measurement, the refractometer was calibrated with distilled water. Three small pieces of pulp of about 5 g each were cut out from the ROI. Their liquid content was dropped into a sample tank by manual extrusion. The average value of the solid content of the three pulp samples was taken as the measurement value.

### 2.4. Spectral Preprocessing

The standard normal variable transformation (SNV) is a normalization, which is sometimes employed in near infrared spectroscopy [25,26]. This preprocessing algorithm can center and scale each spectrum. Multiple scatter correction (MSC) is used to compensate for the non-uniform scattering effects in spectral data, when heterogenous sample sizes, irregular distributions, and other physical effects are present [27]. Whereas the Savitsky-Golay (S-G) algorithm can be used to improve smoothness of spectral curves. The different preprocessing effects obtained with MSC, SNV, MSC-SG, and SNV-SG were compared to evaluate the characteristics of the PLSR models.

### 2.5. Division Calibration Set and Validation Set

The sample set partitioning method based on the joint x-y distance algorithm (SPXY) was proposed by Galvão et al. [28]. This algorithm considers the reflection spectrum distribution and the standard value distribution equally important in the data characterization process by increasing the representativity of both the calibration and validation set. In this study, the calibration set and the validation set were grouped by SPXY with a 3:1 ratio.

### 2.6. Selection of Important Wavelengths

The principal components were determined by using the partial least square regression (PLSR) models established via the 5-fold cross-validation method to select the most significant wavelengths in different iterations. The process of selecting the important wavelengths for one quality parameter in Round I is shown in Figure 2.

Here, $Spec_{(I-1)in}$ corresponds to the matrix of the spectral data, which is composed of the set of wavelengths selected during the last iteration, $Y_k$ refers to the measurement value matrix of the $k^{th}$ quality parameter. $N_I$, and $C_I$ in the figure correspond to the binary matrix lines in the $I^{th}$ iteration, and the optimal wavelengths selected in the $(i-1)^{th}$ iteration, respectively, $Num\_total_I$ is the total number of uninformative and interfering wavelengths.

According to the number of wavelengths selected in the $(i-1)^{th}$ iteration, a binary matrix shuffler filter, $M_{Iin}$, with $C_I$ columns and $N_I$ rows for Round I was generated. The value of $M_{Iin}(i,j)$ indicates that Wavelength i is used to construct the predicting quality model j. The root mean square error $RMSECV_{Iin}(:,j)$ for the $N_I$ possible wavelength combinations was calculated separately. Each $RMSECV_{Iin}(:,j)$ value was set as $RMSECV_{Iin}(i,j)$. The binary matrix $M_{Iex}$ was obtained by inverting the elements of $M_{Iin}$, implying a change

in the including state of the sample spectrum for its corresponding wavelength. A new PLSR prediction model and its corresponding root mean square error $RMSECV_{Iex}(i,j)$ was calculated when the inclusion state of wavelength j changed into the $i^{th}$ wavelength.

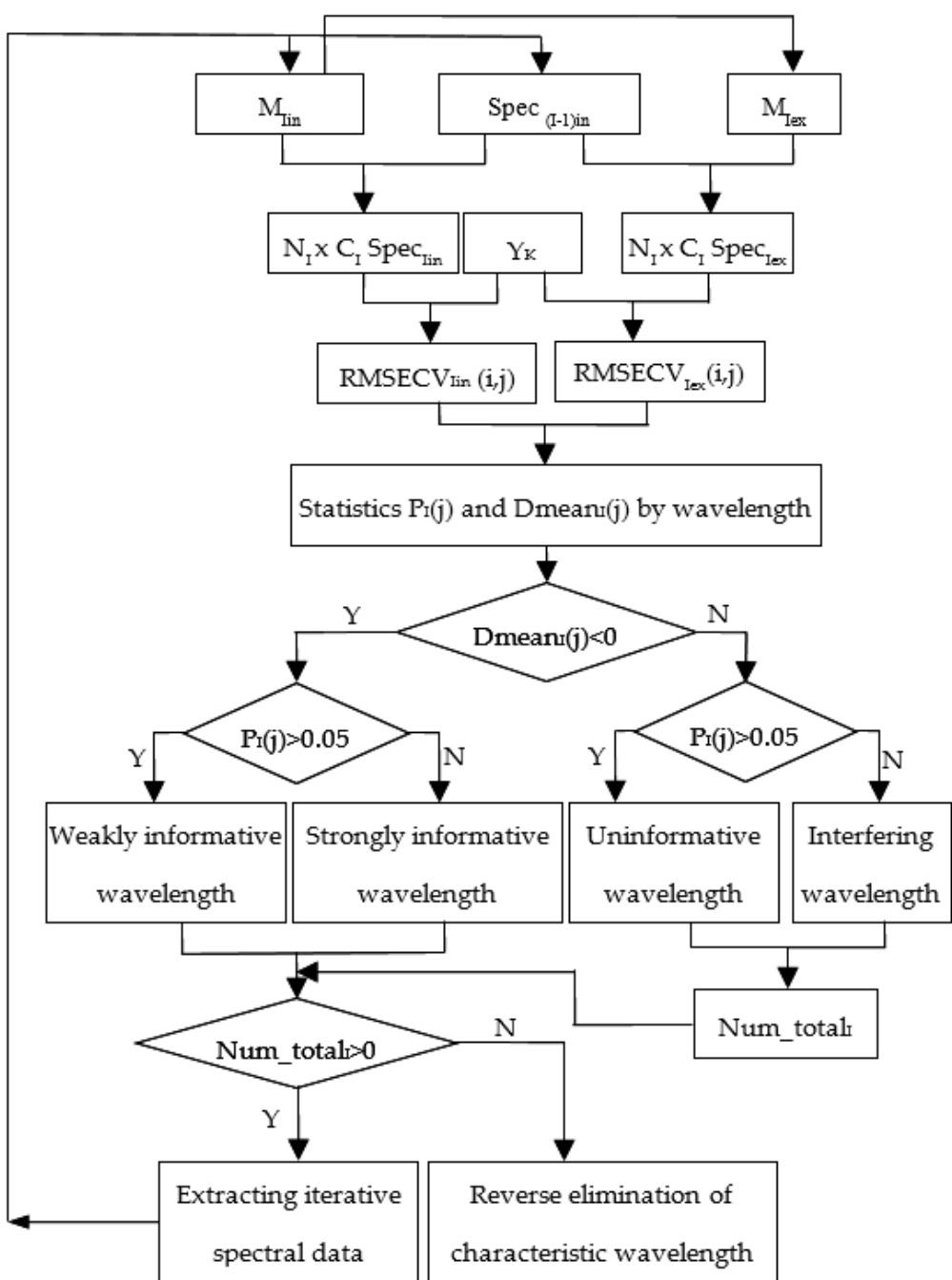

**Figure 2.** Iteration process of Round I.

The values of $RMSECV_{Iin}(i,j)$ and $RMSECV_{Iex}(i,j)$ of the $i^{th}$ wavelength combination with and without including the wavelength j were calculated according to the $M_{Iin}$ and $M_{Iex}$ values. $RMSECV_{Iex}(i,j)$ and $RMSECV_{Iin}(i,j)$ were tested via the Mann–Whitney U test with a significance level of 0.05. The difference between the two values of the wavelength j was defined as $Dmean_I(j)$. The wavelengths were classified into four types with the test level $P_I(j)$ and $Dmean_I(j)$, as shown in Table 1. Strongly informative wavelengths can be used in to drive prediction models, contrarily to weakly informative wavelengths. Interfering

wavelengths create noise inside the model and lower significantly its performance, whereas uninformative wavelengths play the same role of interfering wavelengths but have a lower effect on the model performance.

**Table 1.** Variable classification rules.

| Wavelength Type | Classification Rules |
|---|---|
| Strongly informative wavelength | Dmean(j) < 0, P(j) < 0.05 |
| Weakly informative wavelength | Dmean(j) < 0, P(j) > 0.05 |
| Uninformative wavelength | Dmean(j) > 0, P(j) > 0.05 |
| Interfering wavelength | Dmean(j) > 0, P(j) < 0.05 |

When DmeanI(j) was smaller than 0, its corresponding wavelength was entered into a new iteration. When the number of uninformative and interfering wavelengths (Num_totalI) was smaller than 0, the iteration stopped and the RMSECV value was calculated using the spectra with strongly and weakly informative wavelengths together with their quality values.

Reverse elimination was then performed. When either a strongly informative wavelength or a weakly one was eliminated, a new set of PLSR models was established and the corresponding RMSECV' values were obtained. If the RMSECV' was smaller than the RMSECV, the corresponding wavelength was eliminated and remaining wavelengths were defined as important ones.

### 2.7. Modeling Algorithm

The least square support vector machine (LS-SVM) is an improved SVM algorithm proposed by Suykens [29]. Its operation speed can be significantly improved by solving a set of linear equations instead of the complex quadratic programming problem of the SVM. In this work, the radial basis function (RBF) was used as the kernel function, and the combination of the regression error weight, $\gamma$, and the kernel function parameter, $\sigma^2$, were optimized via grid search based on the cross-validation model. The quality parameters of the LS-SVM models were evaluated by using the $RMSE_C$, $R_C$, $RMSE_V$, and $R_V$ values. The results show that the model performs better when $RMSE_C$ and $RMSE_V$ are small and $R_C$ and $R_V$ are large.

## 3. Results

### 3.1. Statistics and Analysis of the Sensory and Physicochemical Values

The statistic values of a*, the firmness, and the SSC of Korla fragrant pears are shown in Table 2. The value of a* lies in the −7.108–3.254 range. When a* is positive the color of the tested area is red, whereas when a* is negative is green. The firmness lies in the $10.4 \times 10^5$–$14.1 \times 10^5$ Pa range. This value is larger than that measured in other studies [30,31] probably because, in this work, the skin of the pears was not removed. This method was preferred since it meets the most common eating habits of the customers, who generally eat the pears with the skin to increase their uptake of vitamin C. The SSC lies in the 10.0–13.4 °Brix range. Such range is narrower that the one defined by Yu X J et al. and Li J B et al. probably due to the differences in planting locations. On the other hand, the value ranges in the calibration set include those in the other set: Both sets, in fact, are representative since the mean values and dispersion degree of the two sets are similar.

**Table 2.** Statistics of the quality parameters in the calibration and validation sets.

| Quality Parameters | Group | Min | Max | Mean Value | Standard Deviation |
|---|---|---|---|---|---|
| a* | Correction set | −7.108 | 3.254 | −3.459 | 0.987 |
| | Verification set | −5.794 | 2.282 | −3.989 | 0.997 |
| Firmness ($10^5$ Pa) | Correction set | 10.4 | 14.1 | 12.1 | 0.760 |
| | Verification set | 10.8 | 13.4 | 11.3 | 0.637 |
| SSC (°Brix) | Correction set | 10.0 | 13.4 | 12.1 | 0.693 |
| | Verification set | 11.5 | 13.2 | 12.2 | 0.443 |

The a* color space method recommended by the International Commission on illumination (CIE) used "*" in the expression of three parameters.

### 3.2. Spectrum Data Processing

### 3.2.1. Spectral Curves

The spectral curves with the largest distance in most wavelengths are shown in Figure 3. The measured values of a*, of the firmness, and of the SSC of sample 1 and sample 2 are 3.194, $13.9 \times 10^5$ Pa, and 12.0 °Brix and −6.934, $10.4 \times 10^5$ Pa, and 10.1 °Brix, respectively. Three reflection valleys can be observed near 1140 nm, 1440 nm, and 1640 nm, whereas two reflection peaks are located at 960 nm and 1270 nm. A water absorption band exists near 960 nm [32]. The reflection valleys near 1140 nm and 1640 nm may correspond to the first and second overtones of the C-H group, respectively [33]. The strong reflection valley at 1440 nm can be assigned to the first overtone of the O-H and N-H bonds [34]. The reflection peaks near 1270 nm may be related to the second overtones of the O-H and C-H bonds, respectively [35].

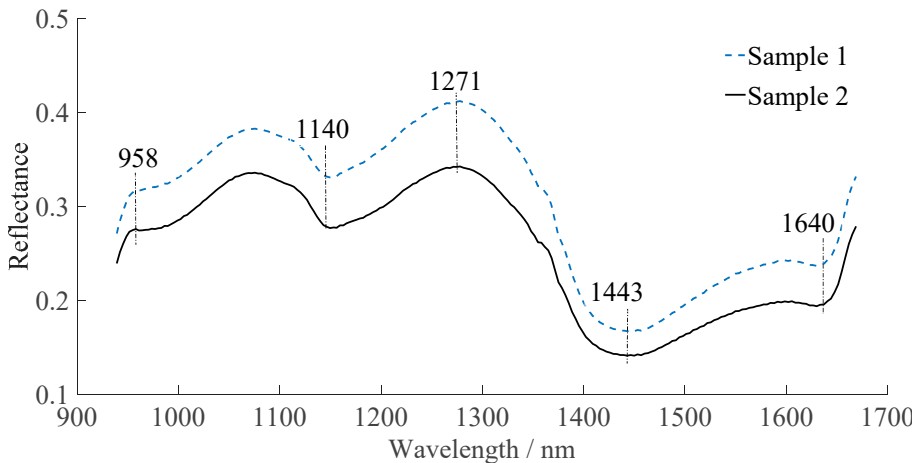

**Figure 3.** Reflective spectral curves.

### 3.2.2. PLSR Models for the Quality Parameters and Optimization of the Principal Components Based on the Full Spectral Analysis

The PLSR models for the a* value, the firmness, and the SSC of Korla fragrant pears were obtained by analyzing the spectral data after different spectral pre-processing processes. The spectra after pretreatment with MSC-SG are shown in Figure 4. The selection process of the numbers of principal components is shown in Figure 5. The principal components to determine the a* value, the firmness, and the SSC are 10, 8, and 9, respectively. The prediction results are listed in Table 3. The results show that the PLSR models with MSC-SG pretreatment exhibit the highest evaluating ability. The $R_C$ and $RMSE_C$ values obtained for a* measure 0.907 and 0.448, respectively, in the case of the calibration set, whereas $R_V$ and $RMSE_V$ measure 0.894 and 0.402 when the validation set is used. The $R_C$ and $RMSE_C$ values of the firmness are 0.914 and $0.352 \times 10^5$ Pa, respectively, for the calibration set and the $R_V$ and $RMSE_V$ values of 0.903 and $0.317 \times 10^5$ Pa, respectively,

are obtained from the validation set. The $R_C$ and $RMSE_C$ of the SSC measure 0.925 and 0.314 °Brix, respectively, when the calibration set is considered, whereas $R_V$ and $RMSE_V$ measure 0.912 and 0.301 °Brix, respectively, in the case of the validation set.

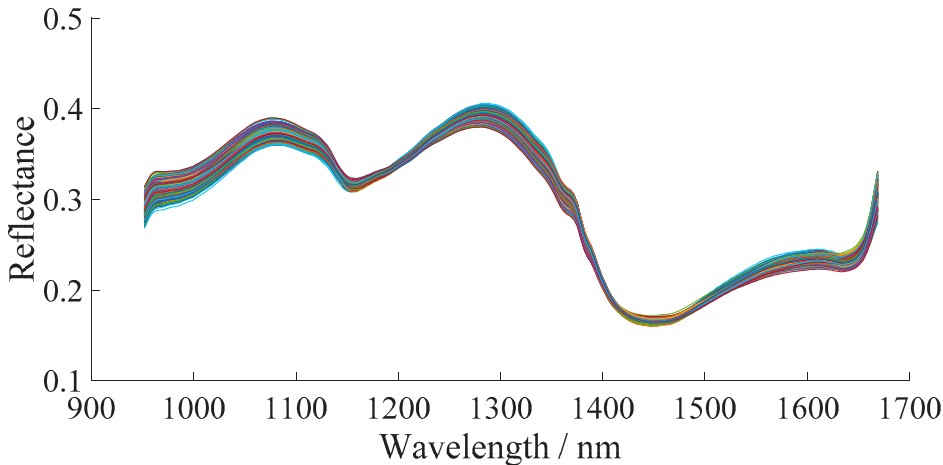

**Figure 4.** Spectra after the MSC-SG preprocessing.

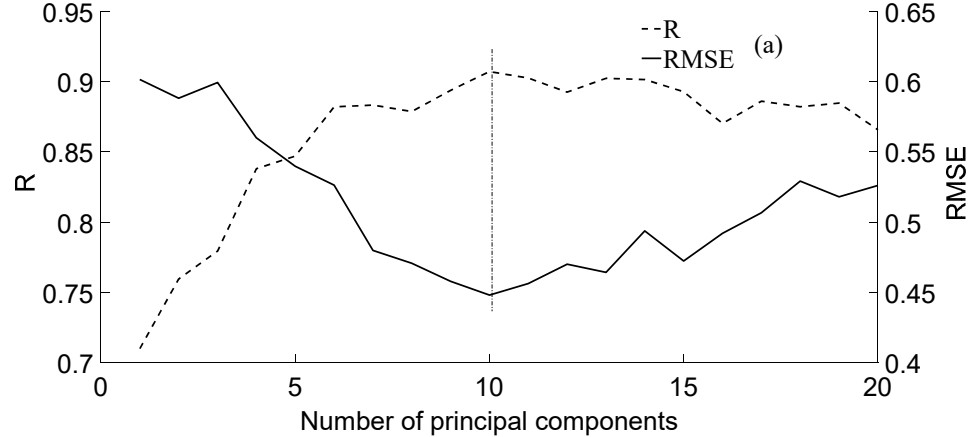

(**a**)

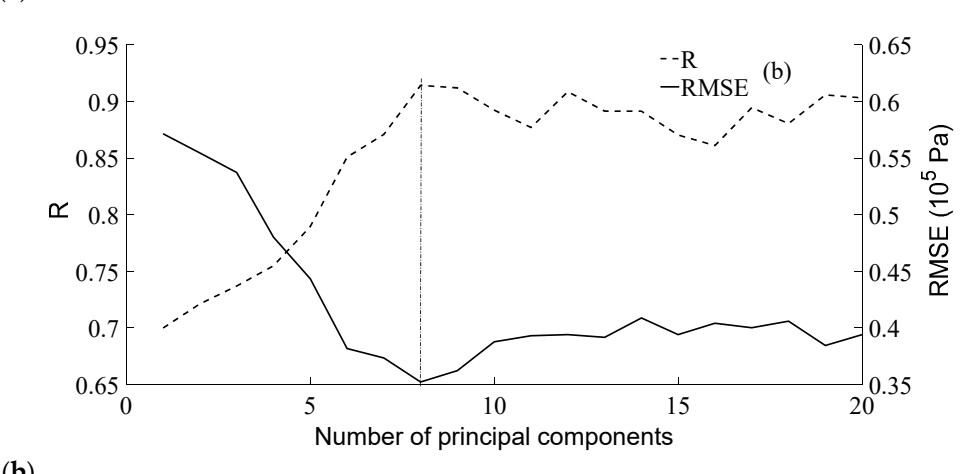

(**b**)

**Figure 5.** *Cont.*

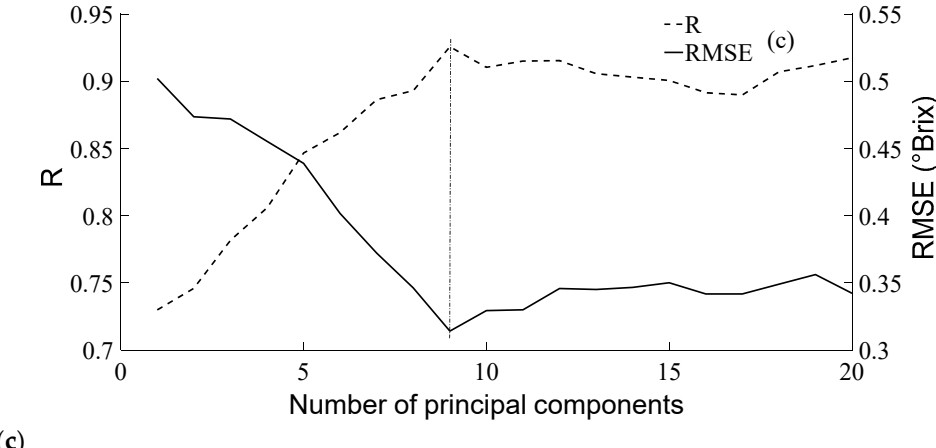

**(c)**

**Figure 5.** Selection process of principal components based on MSC-SG. (**a**) Selecting process to estimate the a* value. (**b**). Selecting process to estimate the firmness. (**c**) Selecting process to estimate the SSC.

**Table 3.** Modeling results to estimate the quality parameters for Korla pears.

| Quality Parameters | Preprocessing Algorithm | $R_C$ | $RMSE_C$ | $R_V$ | $RMSE_V$ |
|---|---|---|---|---|---|
| a* | MSC | 0.875 | 0.552 | 0.867 | 0.522 |
| | SNV | 0.872 | 0.551 | 0.873 | 0.568 |
| | MSC + S − G | 0.907 | 0.448 | 0.894 | 0.402 |
| | SNV + S − G | 0.896 | 0.437 | 0.882 | 0.484 |
| Firmness ($10^5$ Pa) | MSC | 0.892 | 0.357 | 0.898 | 0.338 |
| | SNV | 0.898 | 0.399 | 0.881 | 0.322 |
| | MSC + S − G | 0.914 | 0.352 | 0.903 | 0.317 |
| | SNV + S − G | 0.906 | 0.397 | 0.894 | 0.379 |
| SSC (°Brix) | MSC | 0.914 | 0.410 | 0.903 | 0.480 |
| | SNV | 0.894 | 0.415 | 0.883 | 0.482 |
| | MSC + S − G | 0.925 | 0.314 | 0.912 | 0.301 |
| | SNV + S − G | 0.915 | 0.339 | 0.902 | 0.322 |

The a* color space method recommended by the International Commission on illumination (CIE) used "*" in the expression of three parameters.

3.2.3. Visualization of the Iterative Process and Selection of the Important Wavelengths

In an iterative process, wavelengths can be classified into different groups according to their P and Dmean values. Figure 6 shows the distribution of the P and D-means values for each wavelength obtained in the second iteration. The strongly informative wavelengths, weakly informative wavelengths, uninformative wavelengths, and interfering wavelengths are 7, 57, 37, and 14 to estimate the a* value, 15, 37, 47, and 7 to define the firmness, and 8, 59, 34, and 13 to calculate the SSC, respectively.

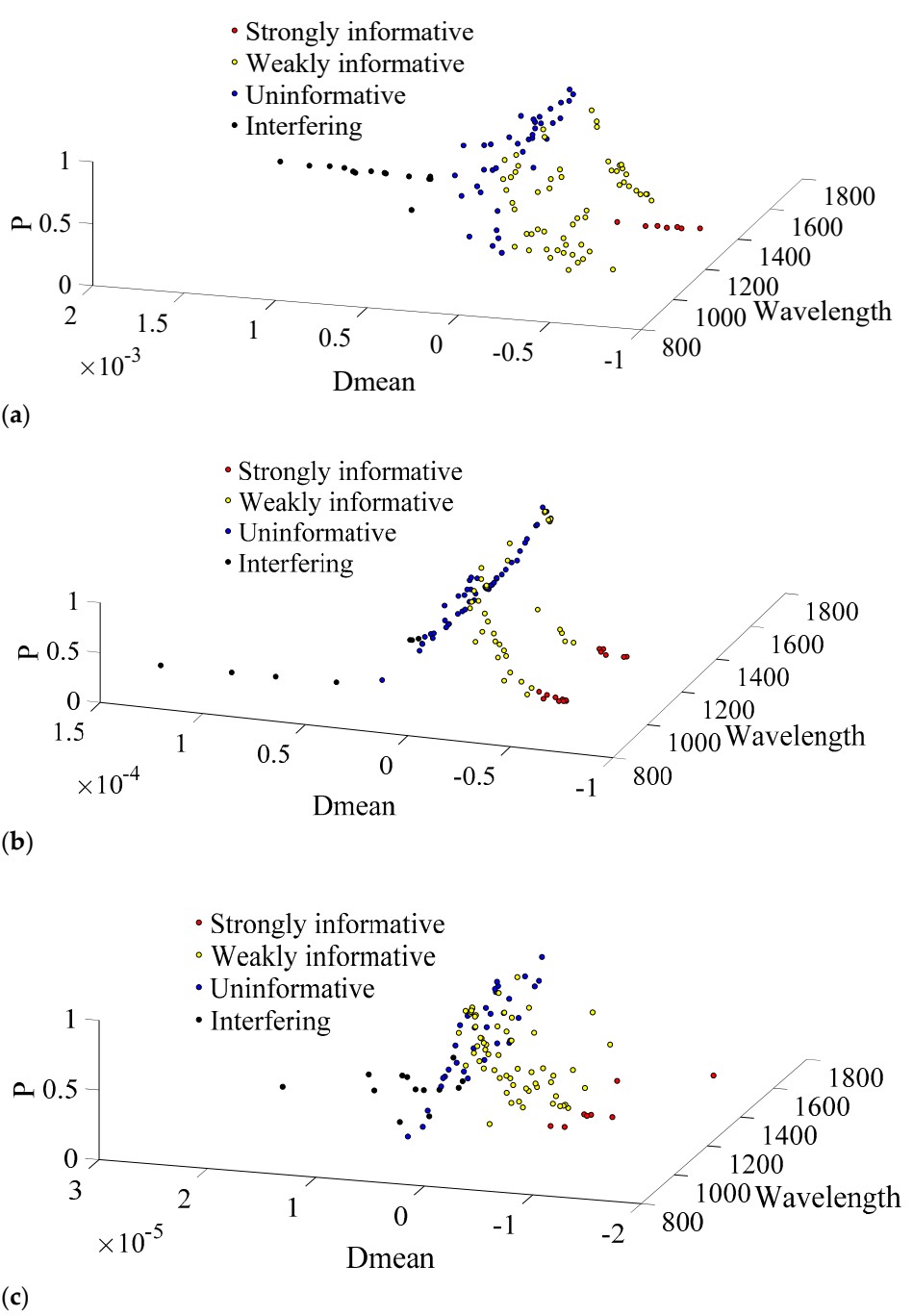

**Figure 6.** Distribution of wavelengths for different parameters obtained in the second iteration. (**a**) Wavelengths to estimate the a* value. (**b**) Wavelengths to estimate the firmness. (**c**) Wavelengths to estimate the SSC.

The number of wavelengths selected for a*, the firmness, and the SSC in different iterations are shown in Figure 7. Their number in the first three rounds initially decreases rapidly and then slows down. Both the irrelevant wavelengths and the interference wavelengths are completely removed after the 6th iteration. The important wavelengths, which were missed during the process, were selected after reverse elimination. To estimate the a* value, the firmness, and the SSC, 8, 11, and 16 important wavelengths are necessary. Selected wavelengths for each parameter are shown in Table 4. The number of important wavelengths of different quality parameters accounts for 3.9%, 5.4%, 7.9% of the valid wavelengths, respectively.

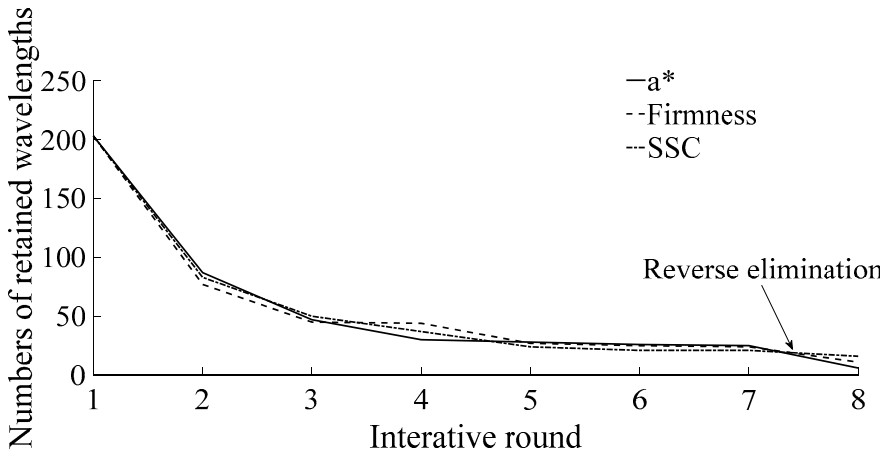

**Figure 7.** Number of retained wavelengths in each IRIV iteration.

**Table 4.** Important wavelengths for different parameters.

| Quality Parameters | Important Wavelengths |
| --- | --- |
| a* | 1078.70 nm, 1130.32 nm, 1238.28 nm, 1321.41 nm, 1453.38 nm, 1508.33 nm, 1535.98 nm, and 1605.63 nm |
| Firmness | 1114.14 nm, 1185.69 nm, 1254.82 nm, 1341.53 nm, 1392.12 nm, 1405.68 nm, 1453.38 nm, 1477.36 nm, 1529.06 nm, 1570.71 nm, and 1616.14 nm |
| SSC | 1046.67 nm, 1053.06 nm, 1179.15 nm, 1211.93 nm, 1234.98 nm, 1241.59 nm, 1304.69 nm, 1385.35 nm, 1415.87 nm, 1463.65 nm, 1487.67 nm, 1491.11 nm, 1508.33 nm, 1518.68 nm, 1581.17 nm, and 1630.18 nm |

The a* color space method recommended by the International Commission on illumination (CIE) used "*" in the expression of three parameters.

### 3.2.4. Evaluation of the Quality Parameters Based on the LS-SVM Model

In this study, several evaluation models were established based on the LS-SVM and the PLSR methods for a set of selected wavelengths. The optimal combinations of the regression error weight, $\gamma$, and the kernel function parameter, $\sigma^2$, are ($8.67 \times 10^4$, $1.21 \times 10^3$), ($1.45 \times 10^4$, $2.93 \times 10^4$), and ($2.37 \times 10^5$, $3.80 \times 10^3$) for the a* value, the firmness, and the SSC, respectively. Figure 8a–c shows the results on the 3 quality parameters obtained via the IRIV-LS-SVM model. The $R_C$ and $R_V$ values measure 0.932 and 0.927, respectively, in the case of the a* value; They are 0.954 and 0.948 for the firmness, and 0.955 and 0.953 for the SSC. The $RMSE_C$ and $RMSE_V$ value measure 0.426 and 0.475, respectively, for the a* value, $0.310 \times 10^5$ Pa and $0.345 \times 10^5$ Pa for the firmness, and 0.319 °Brix and 0.346 °Brix for the SSC.

The principal components used in the PLSR models to estimate a*, the firmness, and the SSC are 8, 8, and 9, respectively. Figure 8d–f shows the results obtained by using the IRIV-PLSR model. The $R_C$ and $R_V$ values of a* measure 0.921 and 0.915, respectively, in the case of the firmness, these values are 0.940 and 0.933, respectively, whereas for the SSC, the measure 0.951 and 0.942. The $RMSE_C$ and the $RMSE_V$ of the a* are 0.447 and 0.406, in the case of the firmness they measure $0.330 \times 10^5$ Pa and $0.395 \times 10^5$ Pa, whereas for the SSC 0.346 °Brix and 0.340 °Brix, respectively.

These results show that the IRIV-LS-SVM model provides more accurate results than the IRIV-PLSR one.

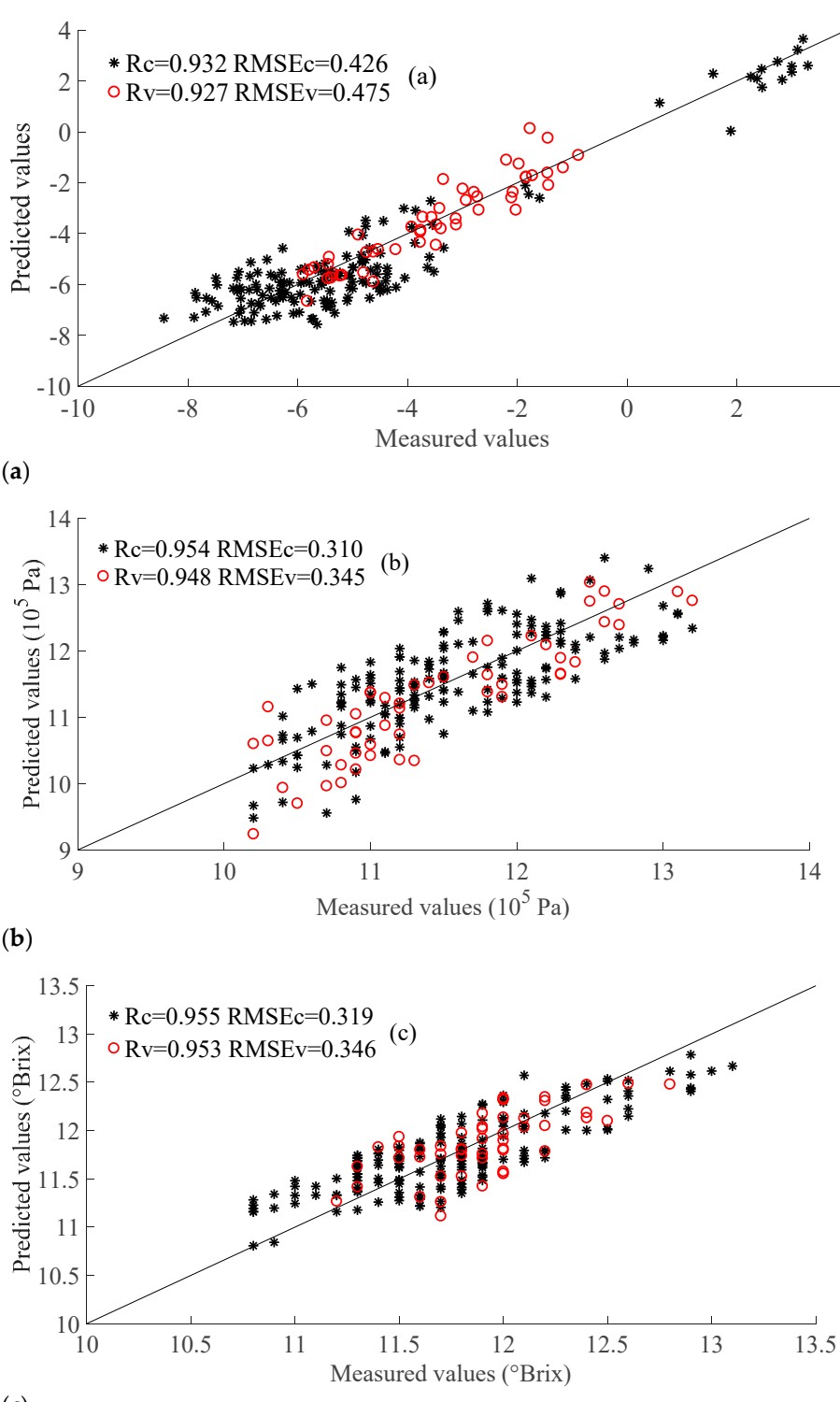

**Figure 8.** *Cont.*

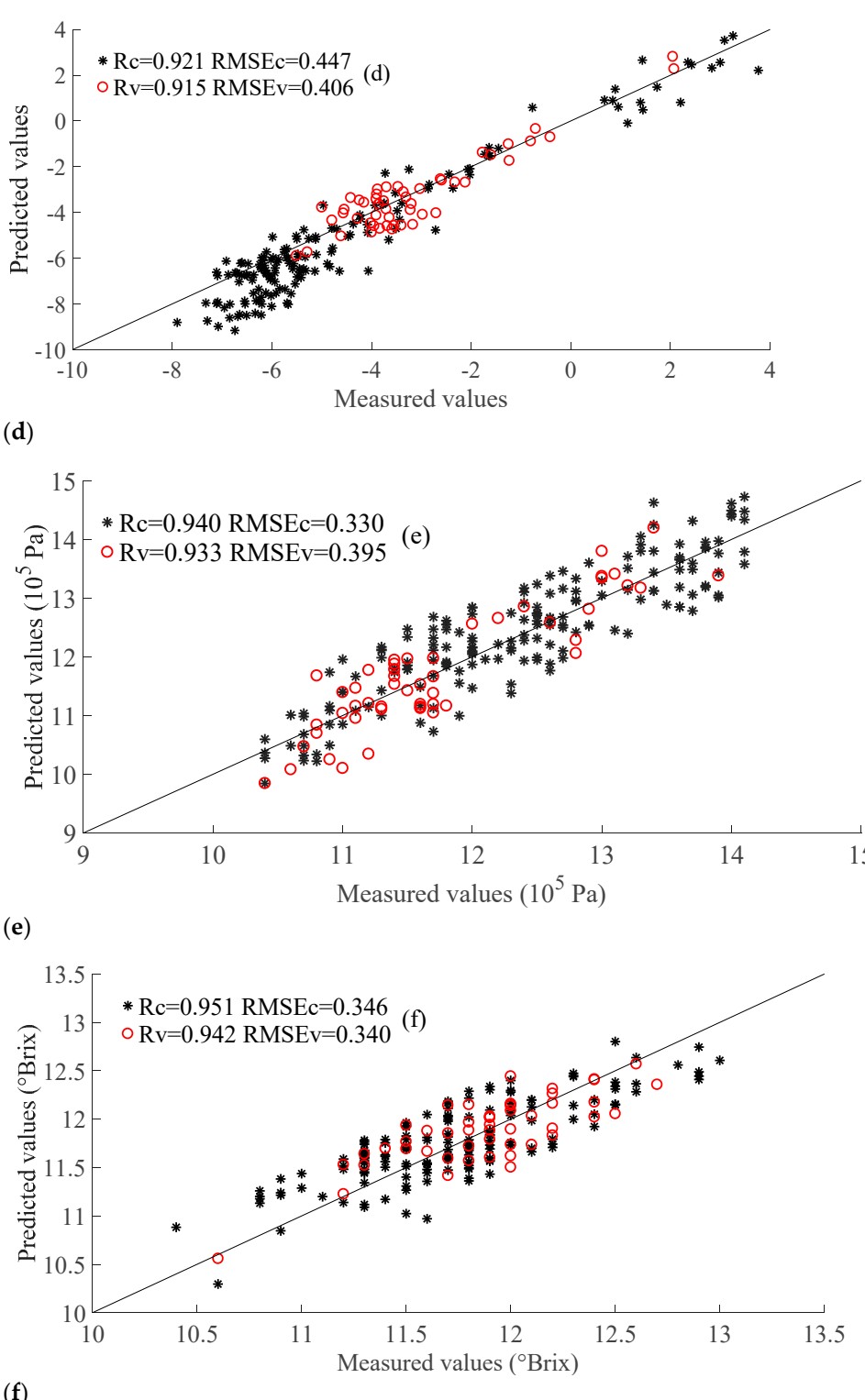

(**d**)

(**e**)

(**f**)

**Figure 8.** Scatter plots of the calibration set (*) and prediction set (o) for each quality parameter. (**a**) Scatter plots of the LS-SVM mold of the a* value. (**b**) Scatter plots of the LS-SVM mold of the firmness. (**c**) Scatter plots of the LS-SVM mold of the SSC. (**d**) Scatter plots of the PLSR mold of the a* value. (**e**) Scatter plots of the PLSR mold of the firmness. (**f**) Scatter plots of the PLSR mold of the SSC.

## 4. Discussion

This work demonstrates that hyperspectral imaging can be used to quantitatively analyze the a* value, the firmness, and the SSC of Korla fragrant pears. Both the PLSR and the LS-SVM models were implemented in combination with the IRIV algorithm to select the important wavelengths. The optimal ($\gamma$ and $\sigma^2$) combinations found in this study are ($8.67 \times 10^4$, $1.21 \times 10^3$), ($1.45 \times 10^4$, $2.93 \times 10^4$), and ($2.37 \times 10^5$, $3.80 \times 10^3$) for the a* value, the firmness, and the SSC, respectively. In the LS-SVM model, the combination of the $R_C$ and $RMSE_C$ values for a*, the firmness, and the SSC measures (0.892, 0.726), (0.914, 0.410), and (0.925, 0.319), respectively. These combinations are (0.883, 0.775), (0.908, 0.548), and (0.916, 0.346), respectively, when the validation set is considered. These results show that the IRIV-LS-SVM model can efficiently evaluate the main important parameters of Korla fragrant pears, which can be used for the quantitative evaluation and grading of fruit.

## 5. Conclusions

Compared with traditional detection methods, multiple parameter detection based on hyperspectral imaging technology has the technical advantages of being nondestructive, real-time and accurate.

There were two ways to reduce the spectral influences caused by different optical path lengths of ROI of Korla fragrant pear. Firstly, there were four halogen light sources at the same vertical plane in the irreflexive hyperspectral imaging system. The center of the four lights was in the center of the moving stage. Secondly, some spectral preprocessing algorithms were used in order to reduce the effects. The combination of MSC and SG exhibited the highest evaluating ability.

Most previous studies predicted only one or two parameters of fruits by non-destructive technologies. Three quality parameters related to the maturity and grading were predicted at the same time in this paper. Both the PLSR and the LS-SVM models were implemented in combination with the IRIV algorithm to select the important wavelengths. Both the irrelevant wavelengths and the interference wavelengths are completely removed after the 6th iteration. 8, 11, and 16 important wavelengths are selected to estimate the a* value, the firmness, and the SSC. The optimal ($\gamma$ and $\sigma^2$) combinations found in this study are ($8.67 \times 10^4$, $1.21 \times 10^3$), ($1.45 \times 10^4$, $2.93 \times 10^4$), and ($2.37 \times 10^5$, $3.80 \times 10^3$) for the a* value, the firmness, and the SSC, respectively. In the LS-SVM model, the combination of the $R_C$ and $RMSE_C$ values for a*, the firmness, and the SSC measures (0.892, 0.726), (0.914, 0.410), and (0.925, 0.319), respectively. These combinations are (0.883, 0.775), (0.908, 0.548), and (0.916, 0.346), respectively, when the validation set is considered. These results show that the IRIV-LS-SVM model can efficiently evaluate the main important parameters of Korla fragrant pears, which can be used for a quantitative evaluation and grading of the fruit. At the same time, this study also has a certain guiding significance for the qualitative detection of other fruits.

There are some research demands in the future. Firstly, a large number of experiments are needed to extend this method to more fruit detection fields through the adjustment of key parameters and the development of supporting equipment. Secondly, the number of Korla fragrant pears can be increased, so as to guarantee the grading quality and realize the industrial upgrading. Thirdly, this research mainly used spectral data to quantitatively predict the quality parameters of Korla fragrant pear although hyperspectral imaging technology has the characteristics of atlas integration. The image processing technology can be introduced to identify the kind of defects, defect level, maturities, et al. of Korla fragrant pear according to more organoleptic attributes.

**Author Contributions:** Resources, F.C.; data curation, R.S.; writing—original draft preparation, T.W.; writing—review and editing, Y.L.; visualization, C.H.; supervision, J.C.; project administration, Y.L. All authors have read and agreed to the published version of the manuscript.



**Funding:** This work was supported by the National Natural Science Foundation of China, grant number 31960498, the Open Project Program of the Key Laboratory of Colleges & Universities under the Department of Education of Xinjiang Uygur Autonomous Region, grant number TDNG20200102, and the Science and Technology Planning Project of 1st Division of Xinjiang Production and Construction Corps, grant number 2019XX02.

**Institutional Review Board Statement:** Not applicable.

**Informed Consent Statement:** Not applicable.

**Data Availability Statement:** The data presented in this study are available on request from the corresponding author. The data are not publicly available due to the request of funding scientific research projects.

**Acknowledgments:** This work was supported in part by the National Natural Science Foundation of China (Project No. 31960498), the Open Project Program of the Key Laboratory of Colleges & Universities under the Department of Education of Xinjiang Uygur Autonomous Region (No. TDNG20150101 and No. TDNG20200102), the Science and Technology Planning Project of 1st Division of Xinjiang Production and Construction Corps (No. 2019XX02). The authors are grateful to anonymous reviewers for their comments.

**Conflicts of Interest:** The authors declare that they have no known competing financial interest or personal relationship that could have appeared to influence the work reported in this paper.

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
