# Peer review of "Quantitative Evaluation of Color, Firmness, and Soluble Solid Content of Korla Fragrant Pears via IRIV and LS-SVM"

_agriculture, doi:10.3390/agriculture11080731_

Round 1
Reviewer 1 Report
The paper is very interesting and in the scope of the journal. Some points should be clarified before continuing with the process:
Fig 1. Please correct the word "sample"
2.3 Measurement of the sensory and physicochemical parameters. The term " sensory" didn't seem to be correct. The authors measured the colour a* parameter not performed a sensorial evaluation. Please, replace it here and throughout the text.
How many pears did the authors analyse? Please add information.
Please add information: SI Units of the firmness and SSC parameters
Results section.
Line 224 and 225 - Why did the authors refer to a "statistic values". What was the statistical treatment performed on the data? Please, clarify.
Line 231, please remove the negative sign before the number 13.4
Were the SSC unit in percentage not in ºBrix? Why? Please, clarify.
The paper should have a conclusion section. Please, add information
Reviewer 2 Report
Liu et al. developed a non-invasive tool of monitoring three major pear fruit quality attributes (color, firmness, sugar content). The experiment was well-conducted, and the results are interesting. However, several issues ought to be dealt.
(1) Line 2: no need to give the cultivar name in the title, except the case that the authors believe that the obtained findings are relevant only on that cultivar.
(2) Lines 14/ 34: Although the term “standards of living improve” correct, it is very puzzling. Thus better rephrase it.
(3) Line 22, “The” before the numbers is wrong. Find another way of introducing them.
(4) Lines 26-27: Do you really need to present these values in the abstract? My suggestion is to erase them
(5) Lines 35-36: Yes indeed sensory attributes are very important for consumer. However, is this relevant here? Did the authors measure sensory attributes? If not, why highlight this limitation here?
(6) Line 36, and the whole introduction: NO need to present the cultivar name here, except the case that the authors believe that the obtained findings are relevant only on that cultivar.
(7) Lines 49-50: the acoustic method is irrelevant to the current study. No need to mention.
(8) Lines 79-90: the whole paragraph needs to go out. These studies were focused on completely different subjects. Thus, only mention that the employed method has been effectively used in other plant science fields.
(9) Lines 102-103: Since you do not use it further in the study, why did you label the sun side? Did you take into account the sun side, when doing the measurements?
(10) The Introduction needs to stress the advantages of non-invasive remote monitoring (several arguments are provided in Taheri-Garavand et al., 2021 Acta Physiologia Plantarum 43, 78). The authors also need to mention that the hyperspectral approach requires expensive equipment, trained personnel and complex data analysis (see relevant section in Fanourakis et al., 2021 Agronomy 11, 795).
(11) Line 110: the setting looks similar to the one developed by Bergsträsser et al., 2015 (Plant Methods 11, 1), except from the fact that they used flat samples (leaves). Please mention this earlier study, and highlight the introduced innovation (3D samples).
(12) Line 232: improper citation (introduce year of publication and reference #)
(13) Lines 236-246: no need to mention every number here. Readers can see them in the Table. Shorten it to 2 lines, or erase it.
(14) Figure 3: By showing all the samples, the reader cannot see. Select the two extreme ones and present them in the figure. In a supplementary table, show the (color, firmness, sugar content) data for these two selected ones, so the readers can see the effects.
(15) Lines 305-312: Insert the selected wavelengths in a new Table, and simply refer the number here (like you do in the abstract)
(16) Line 347: One paragraph is not acceptable as discussion section. You need to extent it to at least 1 page. The text as it is written is purely results. Please see below some suggestions:
Innovation ? 3 D samples as compared to flat ones (Bergsträsser et al., 2015)
Advantage: non-invasive, real-time, remote sensing (Taheri-Garavand et al., 2021 Acta Physiologia Plantarum 43, 78)
Disadvantage: expensive equipment, trained personnel and complex data analysis (Fanourakis et al., 2021 Agronomy 11, 795)
What is different as compared to the studies mentioned in the introduction?
Possibilities of using it in the industry?
Possibility of adjustment for other pear cultivars, or other fruit crops?
Possibility of using it to predict harvest data?
Future work: include sensory attributes
Round 2
Reviewer 2 Report
The authors substantially revised their manuscript, thus I propose its acceptance.